# Understanding Reasons for Vaccination Hesitancy and Implementing Effective Countermeasures: An Online Survey of Individuals Unvaccinated against COVID-19

**DOI:** 10.3390/vaccines12050499

**Published:** 2024-05-05

**Authors:** Yurie Kobashi, Makoto Yoshida, Hiroaki Saito, Hiroki Yoshimura, Saori Nonaka, Chika Yamamoto, Tianchen Zhao, Masaharu Tsubokura

**Affiliations:** 1Department of Radiation Health Management, Fukushima Medical University School of Medicine, Fukushima, Fukushima 960-1295, Japan; yurie-s@umin.ac.jp (Y.K.); 110818makoto@gmail.com (M.Y.); h.saito0515@gmail.com (H.S.); hirosuke0624@gmail.com (H.Y.); chika.y.9112@gmail.com (C.Y.); cho1230@fmu.ac.jp (T.Z.); 2Department of Internal Medicine, Serireikai Group Hirata Central Hospital, Ishikawa Country, Fukushima 963-8202, Japan; 3Faculty of Medicine, Teikyo University School of Medicine, Itabashi-ku, Tokyo 173-8605, Japan; 4Department of Internal Medicine, Soma Central Hospital, Soma, Fukushima 976-0016, Japan; 5School of Medicine, Hiroshima University, Hiroshima 739-8511, Japan; 6Research Center for Community Health, Minamisoma Municipal General Hospital, Minamisoma, Fukushima 975-0033, Japan; 7General Incorporated Association for Comprehensive Disaster Health Management Research Institute, Minato-ku, Tokyo 108-0074, Japan

**Keywords:** vaccine hesitancy, COVID-19, attitudes, practice

## Abstract

This online survey of unvaccinated people living in Japan aimed to identify the reasons for declining vaccination and to develop effective countermeasures. We conducted a hierarchical class analysis to classify participants, examine factors influencing their classification, and provide the information they needed about coronavirus disease 2019 (COVID-19) and trusted sources of COVID-19 information for each group. A total of 262 participants were classified into three groups: Group 1 with no specific reason (28 participants, 10.69%); Group 2 with clear concerns about trust in the vaccine (85 participants, 32.44%), and Group 3 with attitudinal barriers, such as distrust of the vaccine and complacency towards COVID-19, and structural barriers, such as vaccination appointments (149 participants, 56.87%). For each group, females tended to be classified in Group 2 more than Group 1 (Odds ratio (OR) [95% confidential intervals (95%CI)] = 1.64 (0.63 to 2.66), *p* = 0.001) and in Group 3 more than Group 1 (OR [95%CI] = 1.16 (0.19 to 2.12), *p* = 0.019). The information that the participants wanted to know about COVID-19 was different among each group (Safety: *p* < 0.001, Efficacy: *p* < 0.001, Genetic effects: *p* < 0.001). Those who did not receive the COVID-19 vaccine also had lower influenza vaccination coverage (8.02%). Additionally, 38 participants (14.50%) were subject to social disadvantages because they had not received the COVID-19 vaccine. Countermeasures should be carefully tailored according to the target population, reasons for hesitancy, and specific context. The findings of this study may help develop individualized countermeasures to address vaccine hesitancy.

## 1. Introduction

Coronavirus disease 2019 (COVID-19), caused by severe acute respiratory syndrome coronavirus 2 (SARS-CoV-2), causes pneumonia and other respiratory diseases. COVID-19 was first confirmed in China in December 2019 and has since spread rapidly worldwide. COVID-19 has had a significant impact on physical and mental health as well as educational and economic aspects. Many efforts have been made to develop vaccines, and some COVID-19 vaccines have been approved by the World Health Organization (WHO) [1]. However, there are various challenges regarding rolling out the COVID-19 vaccination worldwide, including inequalities pertaining to the receipt of and access to COVID-19 vaccines and vaccine hesitancy [2,3]. Vaccine hesitancy has been observed worldwide and is a major obstacle to controlling the COVID-19 pandemic [4]. In fact, the WHO has named vaccine hesitancy one of the top ten threats to global health [5], and being unvaccinated (compared with being vaccinated) was associated with hospitalization or developing symptoms following COVID-19 infection and long-duration symptoms [6]. Higher vaccination uptake is also important for achieving immunity against SARS-CoV-2 variants and reducing the risk of new variant generation [2]. Therefore, it is necessary to gain further insight into vaccine hesitancy among the unvaccinated and increase vaccination rates.

A previous study revealed five core individual-level determinants of vaccine hesitancy: confidence, complacency, convenience/constraints, risk calculation, and collective responsibility [7]. Additionally, ethnicity, work status, religion, politics, sex, age, education, income, and beliefs in conspiracies have been reported to be demographic factors associated with COVID-19 vaccine hesitancy [8,9]. To promote the uptake of vaccines among the unvaccinated, it is important to understand the reasons for declining vaccination and determine the most-trusted sources of information used in their decision making [10]. Concerns about adverse reactions are the most common reason for declining vaccination, although there are often multiple reasons [10,11]. However, little is known about the specific combination of reasons that make the unvaccinated decide to decline receipt of the COVID-19 vaccine.

Japan ranked among the countries with the lowest vaccine confidence in the world in 2019 [12]. In Japan, the first case of COVID-19 was confirmed on January 2020 and resulted in a total of 33,803,572 cases and 74,694 cumulative deaths as of 9 May 2023 [13]. Vaccines such as BNT162b2 (Pfizer Biotech) and mRNA-1273 (Moderna) were administered in February 2021 and May 2021, respectively. The third vaccination program, involving vaccination with BNT162b2 or mRNA-1273, began in December 2021. However, 24,238,462 people (18.8% of the total Japanese population) had never received the COVID-19 vaccine as of 12 July 2023 [14]. In this context, Japan is a suitable area for identifying the combination of reasons that make the unvaccinated refuse to receive the COVID-19 vaccine.

In this study, we conducted an online survey of the unvaccinated people living in Japan to identify the combination of reasons for declining vaccination and to develop effective countermeasures. We conducted a hierarchical class analysis to classify participants according to the combination of reasons for declining vaccination, examined factors influencing their classification, and identified the information the participants wanted to know about COVID-19 and trusted sources of COVID-19 information according to each group.

## 2. Materials and Methods

### 2.1. Study Design

We recruited participants from a panel of a web survey company (Macromill Inc. Tokyo, Japan). The panel consisted of respondents aged 20 years or older who could answer the questionnaire in Japanese. Registration for the panel was voluntary. Respondents to this company’s surveys are offered “points”, depending on the question volume, which can be used to purchase goods and services from partner companies. Between 20 March 2023 and 22 March 2023, we conducted an online survey of people living in Japan. We included participants from all the regions in Japan. Before the questionnaire was administered, we explained the details of the study to the participants. We conducted a screening survey after the participants had agreed to participate in the study and provided informed consent. In the screening survey, the respondents were asked about the number of COVID-19 vaccinations they had received. We included participants who had never received the COVID-19 vaccine. During participant recruitment, we ensured there was an equal distribution of each age group (20 s, 30 s, 40 s, 60 s, and 60 s and above) for both sexes. The participants then completed a questionnaire regarding their reasons for declining vaccination. In total, 262 participants were included in this study. The ethics committees of Hirata Central Hospital approved this study (number 2023-0308-1). Informed consent was obtained from all the participants. This study was financially supported by Fukushima Medical University.

### 2.2. Questionnaire Survey

Previous reports were used to prepare the questionnaire surveys [7,8,15,16,17]. We also considered the opinions of the medical staff, local government staff, and researchers involved in the vaccination process in Fukushima Prefecture [18,19]. The questionnaire items included sex, age, family structure, income, occupation, comorbidities, history of influenza vaccination since 2019, history of SARS-CoV-2 infection, belief in COVID-19 conspiracies, information the participants wanted to know about COVID-19, trusted sources of information on COVID-19, disadvantages of not being vaccinated, and reasons for declining vaccination (Appendix A). A total of 21 questions regarding reasons for declining vaccination, including confidence issues, convenience issues, compliance issues, and individual/social group influences, were investigated using the following Likert scale: 1, not at all applicable; 2, not very applicable; 3, cannot say either way; 4, somewhat true; and 5, very applicable.

### 2.3. Statistical Analysis

Participants’ characteristics were analyzed descriptively, with categorical variables calculated as frequencies and proportions and continuous variables calculated as means and standard deviations. We created a cumulative bar chart of the reasons for declining vaccination, arranged in descending order regarding the proportion of “somewhat true” and “very applicable” responses. To identify patterns of reasons for declining vaccination, we conducted a hierarchical class analysis using the Ward method to classify participants according to the reasons for declining vaccination. The distance between the data was measured using Euclidean distance, and the variables selected for hierarchical cluster analysis were standardized. The importance of the selected variables was evaluated using random forest with a mean decrease in Gini, 100 decision trees, and unlimited depth. Effect sizes were calculated using Eta-squared (η^2^) test. Steps of the hierarchical cluster analysis were performed using the scikit-learn (Version: 1.2.2), NumPy (Version: 1.22.4), and pandas (Version: 1.5.3), and η^2^ test was performed using the pingouin (Version: 0.5.3) Python (Version: 3.10.12). We conducted a Kruskal–Wallis test on the reasons for declining vaccination to compare the cluster groups. A multinominal logistic regression model was constructed to compare the participant characteristics between Groups 1 and 2 and between Groups 1 and 3. Only age, sex, and the presence or absence of a comorbidity were included as independent variables because of the small sample size used in this study. The variables selected for the multinomial logistic regression analysis were those that were determined to be statistically significant in the univariate multinomial logistic regression analysis. We summarized the information the participants wanted to know about COVID-19 and the trusted sources of COVID-19 information according to each group, and the chi-squared test was conducted. Statistical significance was set at *p* < 0.05. All statistical analyses were performed using STATA/IC (version 15; Lightstone, DL, College Station, TX, USA) and Python (Version: 3.10.12).

## 3. Results

### 3.1. Characteristics of the Participants

Of the 262 participants, 126 (48.09%) were women, with an average age of 48.01 years. Among them, 21 (8.02%) had received the influenza vaccine in 2019, 88 (33.59%) believed in conspiracy theories about COVID-19, and 24 (9.16%) were infected with SARS-CoV-2 (Table 1). The main reasons for declining vaccination were “Concerns about adverse reactions”, “Concerns about vaccine safety”, “Concerns about long-term effects of the vaccine”, and “Distrust of vaccine development and regulators” (Figure 1). A total of 16 (6.11%) participants were discriminated against for not getting vaccinated, 11 (4.20%) were forced to get vaccinated, and two (0.76%) were fired from their jobs.

We created a cumulative bar chart of the reasons for declining vaccination, arranged in the order from “very applicable” to “not at all applicable” responses.

### 3.2. Classification According to Reasons for Declining Vaccination

We conducted a hierarchical class analysis to classify the participants according to the reasons for declining vaccination. As shown in the hierarchical clustering tree (Figure 2A), 262 participants were separated into three distinct groups. The numbers of participants in Groups 1, 2, and 3 were 28 (10.69%), 85 (32.44%), and 149 (56.8%), respectively. The importance of the 21 selected variables was ranked using the random forest algorithm according to the mean decrease in Gini (Figure 2B). The top ten most important variables were “Booking is troublesome”, “Convenience to the venue is not good”, “Busy”, “Concerns about vaccine safety”, “No infection with COVID-19”, “Concerned about long-term effects of the vaccine”, “Information from the government cannot be trusted”, “Concerns about adverse reactions”, “COVID-19 infection does not cause severe illness”, and “Few people around me are vaccinated.”

### 3.3. Comparison of Reasons for Declining Vaccination and Basic Characteristics among the Groups

To compare patterns of the reasons for declining vaccination, means, variances, and effect sizes of the reasons were calculated per group based on the importance of the variables (Table 2). Group 1 had no variables with mean values > 3. Group 2 had mean values above 3 for the following variables: “Concerns about vaccine safety” (mean value, 4.54; effect size, 0.452), “Concerns about long-term effects of the vaccine” (mean value, 4.05; effect size, 0.370), “Information from the government cannot be trusted” (mean value, 3.68; effect size, 0.340), “Concerns about adverse reactions” (mean value, 4.65; effect size, 0.434), and “Distrust of vaccine development and regulators” (mean value, 3.73; effect size, 0.319). Group 3 had mean values above 3 for the following variables: “Booking is troublesome” (mean value, 3.52; effect size, 0.379), “Concerns about vaccine safety” (mean value, 4.36; effect size, 0.452), “Concerns about long-term effects of the vaccine” (mean value, 4.02; effect size, 0.370), “Information from the government cannot be trusted” (mean value, 3.94; effect size, 0.340), “Concerns about adverse reactions” (mean value, 4.41; effect size, 0.434), “COVID-19 infection does not cause severe illness”, (mean value, 3.28; effect size, 0.223) “Distrust of vaccine development and regulators (mean value, 3.95; effect size, 0.319), “Vaccine ineffectiveness” (mean value, 3.59; effect size, 0.206), “Other infection control measures” (mean value, 3.24; effect size, 0.212)”, “Distrust of healthcare in general” (mean value, 3.41; effect size, 0.225), and “Few people around me have been infected with COVID-19” (mean value, 3.26; effect size, 0.181). Group 1 had significantly more males than all the other groups and had younger participants than those in Group 2. Group 3 had a significantly higher proportion of individuals with comorbidities than Group 1 (Table 3).

### 3.4. Information The Participants Wanted to Know about COVID-19 and Trusted Sources of COVID-19 Information for Each Group

In Group 1, eighteen (64.3%) participants stated that there was no specific information they wanted to know, with television being the most trusted source of information (six respondents, 21.4%), followed by public health centers (five respondents, 17.9%), primary care physicians (five respondents, 17.9%), and family (five respondents, 17.9%). The information that Group 2 wanted pertained to safety (58 respondents, 68.2%), adverse reactions (54 respondents, 63.5%), efficacy (34 respondents, 40.0%), and genetic effects (33 respondents, 38.8%). The sources of information trusted by Group 2 were friends (25 respondents, 29.4%), family (24 respondents, 28.2%), healthcare professionals (24 respondents, 28.2%), primary care physicians (20 respondents, 23.5%), and public health centers (20 respondents, 23.5%). Group 3 wanted to know about adverse reactions (80 respondents, 53.7%) and safety (78, 52.3%). The sources of information trusted by Group 3 were family (45 respondents, 30.2%), television (42 respondents, 28.2%), YouTube (37 respondents, 24.8%), healthcare professionals (34 respondents, 22.8%), and Twitter (33 respondents, 22.1%) (Table 4).

## 4. Discussion

This study aimed to identify the reasons for declining vaccination and to develop effective countermeasures among the unvaccinated. Based on the reasons for declining vaccination, the participants were classified into three groups: a group with no specific reason (Group 1), a group with clear concerns about trust in the vaccine (Group 2), and a group with attitudinal barriers, such as distrust of the vaccine and complacency towards COVID-19, and structural barriers, such as vaccination appointments (Group 3). The characteristics of the participants, the information they wanted to know about COVID-19, and their trusted sources of COVID-19 information differed among the groups.

The group with no specific reason for the decline in COVID-19 vaccination mainly consisted of young men. Group 1 had no variables with high mean values for the reasons for declining vaccination. The multinomial logistic regression results revealed that Group 1 had significantly more males than the other groups and had younger participants than Group 2. In Group 1, 18 (64.3%) participants stated that there was no particular information they wanted to know and that they did not have any particularly trusted sources of COVID-19 information. To the best of our knowledge, this is the first time such a group has been identified. As dialog-based interventions are the most effective strategies for responding to issues of vaccine hesitancy [20], efforts should be made to better understand the backgrounds of these participants and to identify effective communication methods.

The group with clear concerns about trust in the vaccine mainly consisted of older adults and women. Group 2 had high mean values for the reasons for declining vaccination for the following variables: “Concerns about vaccine safety”, “Concerns about long-term effects of the vaccine”, and “Concerns about adverse reactions.” They mainly wanted to know about safety, adverse reactions, and efficacy. The sources of information were friends, family, healthcare professionals, primary care physicians, and public health centers. A total of 28 individuals (31.8% of Group 2) believed in conspiracy theories about COVID-19. This is consistent with previous reports that healthcare workers are considered a trustworthy source of information for addressing concerns about safety and gaining knowledge about vaccines [21,22]. For these participants, educational interventions and seminars held by healthcare workers might be effective in sharing necessary information about COVID-19 vaccines’ safety, especially focusing on long-term effects, adverse reactions, and vaccine development history.

The group with attitudinal barriers, such as distrust of the vaccine and complacency towards COVID-19, and structural barriers, such as vaccination appointments, was Group 3. They had high mean values for the following reasons for declining vaccination: “Booking is troublesome”, “Concerns about vaccine safety”, “Concerns about long-term effects of the vaccine”, “Information from the government cannot be trusted”, “Concerns about adverse reactions”, “Distrust of vaccine development and regulators”, and “Vaccine ineffectiveness.” These participants mainly wanted to know about the adverse reactions and safety of COVID-19 vaccines. The sources of information trusted by Group 3 were family, television, YouTube, healthcare professionals, and Twitter. This is consistent with previous reports of there being multiple reasons for vaccination decline [10,11]. This result is also consistent with past studies that revealed that people who hesitate to get vaccinated believe that social media, people around them, and medical professionals are trusted sources of information [21,22,23]. Multicomponent intervention is necessary to address vaccine hesitancy when multiple factors are involved [24]. Countermeasures should be carefully tailored according to the target population, reasons for hesitancy, and specific context [20]. For these participants, educational interventions and seminars about vaccination safety should be performed by providing information about convenient booking using television, online media, and SNS.

Those who did not receive a COVID-19 vaccine also had lower influenza vaccination coverage. Of the 262 participants, only 21 (8.02%) had received the influenza vaccine in 2019, and this proportion was particularly low among those in Group 1 (1 participants, 0.4%). This result was lower than the overall Japanese influenza vaccination coverage for the 2019 season (with an average of 38% for all age groups and 37% for those over 13 years old) [25]. Previous studies have reported that influenza vaccination coverage is associated with COVID-19 vaccine coverage [26]. Perceptions and attitudes towards health-related behaviors, including vaccination, showed some patterns in a past study [25]. Further research may be needed to clarify the association between COVID-19 vaccine hesitancy and other health-related behaviors (such as dietary habits, participation in health check-ups, and other vaccinations). In addition, vaccine hesitancy caused a decrease in herd immunity. COVID-19 infection is a huge problem among immunocompromised hosts because of the high mortality and prolonged infection it generates. The occurrence of a novel variant of COVID-19 was reported among immunocompromised hosts with prolonged COVID-19 infections [27,28]. To mitigate these issues, comprehensive vaccination of the aging population and immunocompromised population should be achieved.

Some participants were subject to social disadvantages because they had not received the COVID-19 vaccine. Among the 262 participants, 16 (6.11%) were discriminated against for not getting vaccinated, 11 (4.20%) were forced to get vaccinated, and two (0.76%) were fired from their jobs. These problems also exist for influenza vaccination [29]. Promoting vaccination by ignoring the views of those who are hesitant to get vaccinated further marginalizes this group [11]. These results should be reflected in the development of measures to improve vaccination uptake.

This study has some limitations that should be considered when interpreting the results. First, the study participants were people living in Japan. Therefore, caution is needed when generalizing and applying the results to other regions for economic and religious reasons. Second, only those who were not vaccinated were included in this study. It is difficult to gain a comprehensive understanding of vaccine hesitancy from this study alone, as it is known that some people get vaccinated despite their hesitancy towards the COVID-19 vaccine [30]. Therefore, it is desirable to conduct similar studies on vaccinated individuals. Third, attitudes toward vaccines change over time; therefore, the results of this study may not be valid after a certain period of time [31]. Fourth, the participants included in the Macromill panel may have been more willing to join to this study than the general population. Therefore, our findings may not be representative of the entire Japanese population. Fifth, the participants enrolled in this survey are not representative of the geographical structure of the Japanese population. Sixth, the proportion of unemployed individuals was higher than the proportion among the general Japanese population. Seventh, the information on COVID-19 status was obtained from self-reported questionnaires. Therefore, asymptomatic individuals could not be identified. Finally, the present study may have issues regarding reliability and validity. However, a questionnaire used in previous studies included in literature reviews was used in this study to ensure reliability. The answers to the questionnaire were based on a Likert scale. This study involved an online survey and used a small sample size. Future research should focus on increasing the sample size and obtaining the results through a face-to-face survey. Regarding validity, this study recruited participants evenly from all over Japan; however, the study population was registered with a specific survey company. Therefore, a representative sample-gathering approach should be applied in future research. Despite these limitations, this study is the first to investigate the patterns regarding the reasons for declining vaccination.

## 5. Conclusions

Based on the reasons for declining vaccination, the participants were classified into three groups: a group with no specific reason, a group with clear concerns about trust in the vaccine, and a group with attitudinal barriers, such as distrust of the vaccine and complacency towards COVID-19, and structural barriers, such as vaccination appointments. The characteristics of the participants, information they wanted to know about COVID-19, and the trusted sources of COVID-19 differed among the groups. To improve vaccination uptake, identifying the reasons for declining vaccination is necessary, and individualized measures should be taken to address them.

## Figures and Tables

**Figure 1 vaccines-12-00499-f001:**
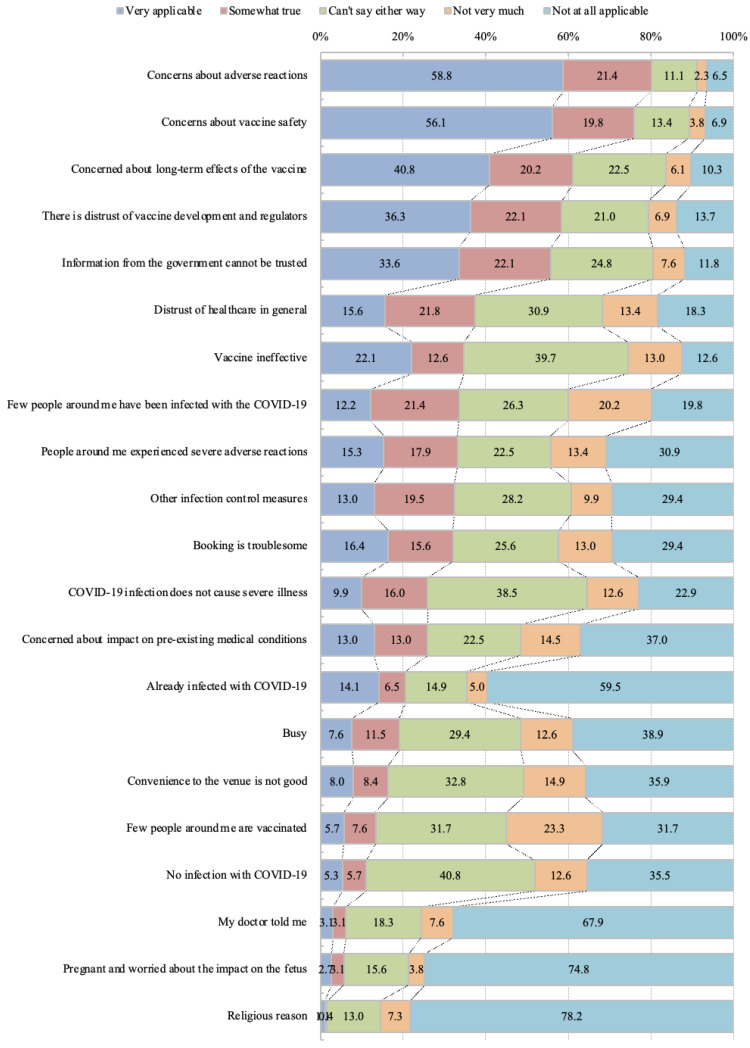
Reasons for declining COVID-19 vaccination.

**Figure 2 vaccines-12-00499-f002:**
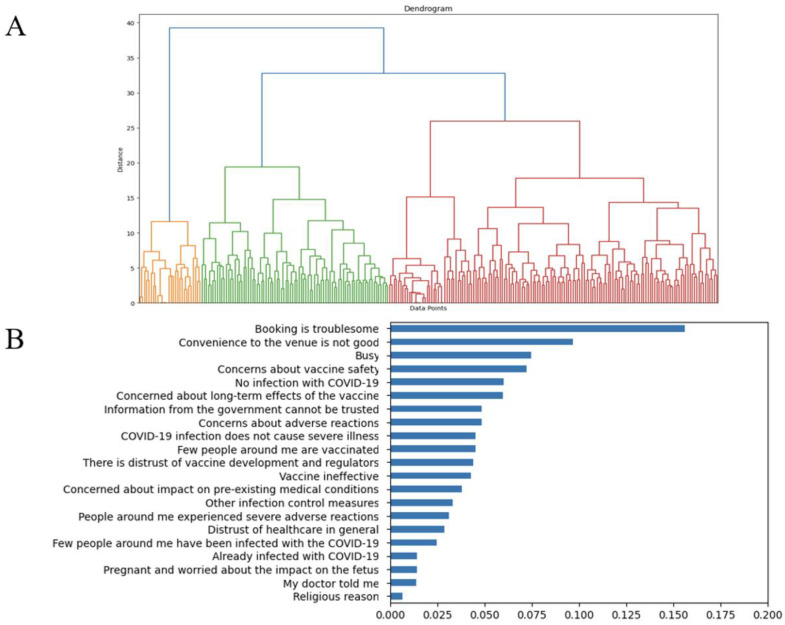
(**A**) Dendrogram of hierarchical cluster analysis based on the reasons for declining COVID-19 vaccination and (**B**) importance of variables used for cluster analysis. (**A**) To identify patterns of the reasons for declining vaccination, we conducted a hierarchical class analysis to classify participants according to the reasons they provided. The variables selected for the hierarchical cluster analysis were standardized. Hierarchical cluster analysis was performed using the Ward method, and the distance between the data was measured using Euclidean distance. (**B**) The importance of the selected variables was evaluated using random forest with a mean decrease in Gini.

**Table 1 vaccines-12-00499-t001:** Basic characteristics of participants.

Variable (N = 262)	n (%)
Female	126 (48.09)
Age (mean [SD])	48.01 (15.09)
Marriage	108 (41.22)
Have Children	118 (45.04)
Family income	
Less than JPY 6 million	153 (58.40)
More than JPY 6 million	56 (21.37)
Do not know	52 (19.92)
Private income	
Less than JPY 2 million	112 (43.08)
More than JPY 2 million	106 (40.46)
Do not know	41 (15.77)
Occupation	
Unemployed	58 (22.14)
Part-time job	42 (16.03)
Housewife/househusband	39 (14.89)
Company employee	77 (29.39)
Self-employed or freelancer	29 (11.07)
Student	6 (2.29)
Other	11 (4.20)
Comorbidity	
Hypertension	29 (11.07)
Diabetes mellitus	11 (4.2)
Bronchial asthma	7 (2.67)
Anaphylactic shock	0 (0.00)
Psychiatric disorders	8 (3.05)
Gout	4 (1.53)
Lipid disorders	6 (2.29)
Rheumatoid arthritis	3 (1.15)
Respiratory diseases	4 (1.53)
Cardiovascular diseases	7 (2.67)
Allergic diseases	15 (5.73)
Immunodeficiency diseases	2 (0.76)
Malignant tumors	5 (1.91)
Thyroid diseases	1 (0.38)
Liver disease	2 (0.76)
Kidney disease	2 (0.76)
Influenza vaccination after 2019	21 (8.02)
Infected with SARS-CoV-2	67 (25.57)
COVID-19 conspiracy beliefs	88 (33.59)
Big pharma is encouraging the spread of coronavirus to make money	71 (27.10)
Coronavirus was developed by the government as part of a bioweapons program	31 (11.83)
5G is causing the coronavirus	14 (5.34)
The coronavirus is a myth to force vaccinations on people	42 (16.03)
There is no such thing as coronavirus	24 (9.16)
What information do you want to know about the COVID-19 vaccine?	
Adverse Reactions	137 (52.29)
Safety	139 (53.05)
Efficacy	83 (31.68)
Genetic Effects	85 (32.44)
Cost	48 (18.32)
Future COVID-19 vaccination schedule	29 (11.07)
None in particular	83 (31.68)
Disadvantages of not being vaccinated	
Discriminated against for not getting vaccinated	16 (6.11)
Forced to vaccinate	11 (4.2)
I was fired from my job	2 (0.76)
Infected with SARS-CoV-2	9 (3.44)
Other	7 (2.67)
None in particular	224 (85.5)

According to the exchange rate, as of 2 August 2023, JPY 1 was equal to USD 0.0070.

**Table 2 vaccines-12-00499-t002:** Reasons for declining COVID-19 vaccination, grouped by variable importance.

	Group 1 (n = 28)		Group 2 (n = 85)		Group 3 (n = 149)			
	Mean	Variance	Mean	Variance	Mean	Variance	*p*-Value	η^2^
Booking is troublesome.	2.32	2.52	1.6	0.77	3.52 *	1.37	<0.001	0.379
Convenience of travelling to the venue is not good.	1.64	0.9	1.45	0.54	3.05	1.32	<0.001	0.371
Busy.	1.64	1.28	1.52	0.9	2.98	1.37	<0.001	0.297
Concerns about vaccine safety.	1.82	1.34	4.54 *	0.8	4.36 *	0.7	<0.001	0.452
No infection with COVID-19.	1.86	1.24	1.64	0.9	2.81	1.13	<0.001	0.229
Concerns about long-term effects of the vaccine.	1.43	0.33	4.05 *	1.57	4.02 *	0.99	<0.001	0.370
Information from the government cannot be trusted.	1.36	0.31	3.68 *	1.98	3.94 *	0.89	<0.001	0.340
Concerns about adverse reactions.	2.07	1.77	4.65 *	0.47	4.41 *	0.73	<0.001	0.434
COVID-19 infection does not cause severe illness.	1.79	1.14	2.22	1.49	3.28	1.08	<0.001	0.223
Few people around me are vaccinated.	1.46	0.48	1.82	1.27	2.77	1.1	<0.001	0.203
There is distrust of vaccine development and regulators	1.36	0.39	3.73 *	2.27	3.95 *	0.96	<0.001	0.319
Vaccine ineffectiveness.	1.75	1.01	2.95	1.81	3.59 *	1.05	<0.001	0.206
Concerns about impact on pre-existing medical conditions	1.61	1.06	2.04	2.06	2.94	1.76	<0.001	0.131
Other infection control measures.	1.25	0.19	2.44	2.11	3.24 *	1.47	<0.001	0.212
People around me have experienced severe adverse reactions.	1.46	0.48	2.88	2.58	2.89	1.8	<0.001	0.092
Distrust in healthcare in general.	1.36	0.39	2.92	1.98	3.41 *	1.15	<0.001	0.225
Few people around me have been infected with COVID-19.	1.54	0.85	2.59	1.96	3.26 *	1.17	<0.001	0.181
Already infected with COVID-19.	1.43	0.77	1.94	2.56	2.33	2.26	0.003	0.038
Pregnant and worried about the impact on the fetus.	1.32	0.37	1.11	0.21	1.85	1.48	<0.001	0.112
My doctor told me.	1.32	0.45	1.28	0.66	1.93	1.39	<0.001	0.088
Religious reason.	1.39	0.47	1.06	0.08	1.58	0.93	<0.001	0.084

We used Likert scales to measure the mean and variance of reasons for declining COVID-19 vaccination per group as follows: 1, not at all applicable; 2, not very applicable; 3, cannot say either way; 4, somewhat true; 5, very applicable. * mean value above 3. We conducted a Kruskal–Wallis test. Effect sizes were calculated using the Eta-squared test. Group 1: group with no specific reason, Group 2: group with clear concerns about trust in the vaccine, and Group 3: group with attitudinal barriers and structural barriers.

**Table 3 vaccines-12-00499-t003:** Multinominal logistic regression analysis of differences between the groups.

	Group 1 (Base Group)	Group 2			Group 3		
	n (%) (or Mean [SD])	n (%) (or Mean [SD])	OR (95% CI) in MLRA	*p*-Value in MLRA	n (%) (or Mean [SD])	OR (95% CI) in MLRA	*p*-Value in MLRA
Female	6 (21.4)	50 (58.8)	1.64 (0.63 to 2.66)	**0.001**	70 (47.0)	1.16 (0.19 to 2.12)	**0.019**
Age	42.5 [13.9]	50.5 [13.5]	0.03 (0.00 to 0.06)	**0.047**	47.6 [15.9]	0.02 (−0.01 to 0.04)	0.29
With comorbidity	4 (14.3)	27 (31.8)	0.71 (−0.49 to 1.92)	0.25	53 (35.6)	1.03 (−0.12 to 2.18)	**0.079**

OR = odds ratio; CI = confidence interval; Significant values are in bold.

**Table 4 vaccines-12-00499-t004:** Differences between the groups regarding the information they wanted to know and the media they trust.

Group	1 (n = 28)	2 (n = 85)	3 (n = 149)	*p*-Value
Variable (n (%))				
What information do you want to know about the COVID-19 vaccine?				
Adverse Reactions	3 (10.7)	54 (63.5)	80 (53.7)	**0.05**
Safety	3 (10.7)	58 (68.2)	78 (52.3)	**<0.001**
Efficacy	1 (3.6)	34 (40.0)	48 (32.2)	**<0.001**
Genetic Effects	0 (0.0)	33 (38.8)	52 (34.9)	**<0.001**
Cost	3 (10.7)	17 (20.0)	28 (18.8)	**0.003**
Future coronavirus vaccination schedules	3 (10.7)	12 (14.1)	14 (9.4)	**0.003**
Nothing in particular	18 (64.3)	16 (18.8)	49 (32.9)	**0.041**
Do you trust the information on COVID-19 from the following sources?				
Facebook	2 (7.1)	2 (2.4)	13 (8.7)	**0.009**
Twitter	4 (14.3)	17 (20.0)	33 (22.1)	**0.002**
Instagram	1 (3.6)	8 (9.4)	16 (10.7)	**0.004**
YouTube	4 (14.3)	17 (20.0)	37 (24.8)	**0.004**
Government	4 (14.3)	12 (14.1)	26 (17.4)	**0.001**
Public health center	5 (17.9)	20 (23.5)	29 (19.5)	**0.001**
Television	6 (21.4)	17 (20.0)	42 (28.2)	**0.005**
Newspaper	4 (14.3)	8 (9.4)	28 (18.8)	**0.008**
Weekly magazine	3 (10.7)	6 (7.1)	7 (4.7)	**0.003**
Primary care physician	5 (17.9)	20 (23.5)	32 (21.5)	**<0.001**
Healthcare professional	4 (14.3)	24 (28.2)	34 (22.8)	**0.005**
Pharmaceutical company	5 (17.9)	5 (5.9)	14 (9.4)	**0.007**
Friend	2 (7.1)	25 (29.4)	31 (20.8)	**0.015**
Family	5 (17.9)	24 (28.2)	45 (30.2)	**0.004**

We conducted a chi-square test. Significant values are in bold.

## Data Availability

Data supporting the findings of this study are available from the corresponding author. However, restrictions apply to the availability of these data. Data were obtained and used under the license for the current study and are not publicly available but can be made available upon reasonable request from the corresponding author and with the permission of the Fukushima Medical University School of Medicine.

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
