# Peer review of "Understanding Reasons for Vaccination Hesitancy and Implementing Effective Countermeasures: An Online Survey of Individuals Unvaccinated against COVID-19"

_vaccines, 2024, doi:10.3390/vaccines12050499_

Round 1

Reviewer 1 Report

Comments and Suggestions for Authors

The study is good and accept. However, the manuscript can be improved. My comments are

1. Specific information about COVID-19 and its vaccines do individual groups need to address their concerns, can be adressed more effectively.
2.
What logistical improvements can be made to make the vaccination process more convenient and less stressful for people who are hesitant due to structural barriers, thereby improving vaccination compliance?
3. Based on the categories identified, what is the impact of vaccine hesitancy on long-term public health goals, particularly herd immunity and control of COVID-19 variants, and what proactive measures can be taken to mitigate this impact?

Overall, the study is insightful and engaging, but it could benefit from an increased number of references to enhance its credibility and depth. Furthermore, still can improve better presentation especially in introduction and more specifically in conclusion by discussing the finding their research work in brief.

Author Response

The study is good and accept. However, the manuscript can be improved. My comments areSpecific information about COVID19 and its vaccines do individual groups need to addresstheir concerns, can be adressed more effectively. 

Response: Thank you for reviewing our manuscript. We have revised the manuscript in accordance with the comments. The following are our responses to the reviewer’s comments.

What logistical improvements can be made to make the vaccination process more convenientand less stressful for people who are hesitant due to structural barriers, thereby improvingvaccination compliance?

Response: Thank you for your comment. We thought below logistical improvements can be made to improve vaccination process and the proportion of participantion. We inserted below sentence in discussion section.

“For these participants, educational interventions and seminars by healthcare workers might be effective in sharing necessary information and knowledge about COVID-19 vaccines safety, especially focusing on long-term effects, adverse reactions, and vaccine development history.”

“For these participants, educational interventions and seminars about vaccination safety should be performed with providing information about convenient booking using television, online media, and SNS.”

Based on the categories identified, what is the impact of vaccine hesitancy on long-term public health goals, particularly herd immunity and control of COVID-19 variants, and what proactive measures can be taken to mitigate this impact?

Response: Thank you for your insight. We inserted below sentence in discussion section.

“Vaccine hesitancy caused decreasing of herd immunity; COVID-19 infection was a huge problem among immunocompromised host, because of the high mortality and prolonged infection. The occurrence of novel variant of COVID-19 were reported among immunocompromised host with prolonged COVID-19 infection. To mitigate these issues, comprehensive vaccination among ageing population and immunocompromised population should be achieved.”

Overall, the study is insightful and engaging, but it could benefit from an increased number of references to enhance its credibility and depth.

Response: Thank you for your comment. We added some references.

Weigang, S.; Fuchs, J.; Zimmer, G.; Schnepf, D.; Kern, L.; Beer, J.; Luxenburger, H.; Ankerhold, J.; Falcone, V.; Kemming, J.; et.al. Within-host evolution of SARS-CoV-2 in an immunosuppressed COVID-19 patient as a source of immune escape variants. Nature communications. 2021, 12(1), 6405; DOI: 10.1038/s41467-021-26602-3.

Baang, J. H., Smith, C., Mirabelli, C., Valesano, A. L., Manthei, D. M., Bachman, M. A., Wobus, C. E., Adams, M. Washer, L., Martin, E. T.; et, al. Prolonged severe acute respiratory syndrome coronavirus 2 replication in an immunocompromised patient. The Journal of infectious diseases. 2021, 223(1), 23-27; DOI: 10.1093/infdis/jiaa666.

Furthermore, still can improve better presentation especially in introduction and more specifically in conclusion by discussing the finding their research work in brief. 

Response: Thank you for your insight. We reduced the finding sentence especially in discussion markedly.

Reviewer 2 Report

Comments and Suggestions for Authors

1.      This paper investigates the reasons for vaccination hesitancy against COVID-19, which is conducive to providing a basis for implementing effective countermeasures.

2.      Which cities or regions are the 262 participants in this survey distributed in? Is this study reflecting the situation in Japan as a whole or in a particular city? Is the survey sample representative? The author should provide an explanation.

3.      The paper needs to provide an analysis of the reliability and validity of the survey.

4.      In Table 1, Basic characteristics of participants, some variables such as Family income, Private income, Occupation are classified too finely, resulting in a small number of items. Suggest merging according to major categories appropriately.

5.      Table 3. (base)B (95% CI)OR or HR?

6.      Table 3. Multinominal logistic regression analysis of differences between the groups

How was the statistical analysis conducted? What are the independent variables? How are independent variables assigned values? What factors have been verified or controlled? Why only compared gender, age, and comorbidities? Why not compare the reasons for debilitating COVID-19 vaccination?

7.      The conclusion is not clear enough. The main reasons for vaccine hesitancy in different groups should be clearly identified, and corresponding policy recommendations should be proposed.

8.      Pay attention to standardizing chart formats. As shown in Table 1: "Coronavirus was developed by the government as part of a bioweapon program." and Figures 1 and 2, all reasons are followed by a “.”.

Comments on the Quality of English Language

Moderate editing of English language required.

Author Response

This paper investigates the reasons for vaccination hesitancy against COVID-19, which is conducive to providing a basis for implementing effective countermeasures.

Response: Thank you for reviewing our manuscript. We have revised the manuscript in accordance with the comments. The following are our responses to the reviewer’s comments.

Which cities or regions are the 262 participants in this survey distributed in? Is this study reflecting the situation in Japan as a whole or in a particular city? Is the survey sample representative? The author should provide an explanation.

Response: Thank you for your comment. We included participants from all region in Japan. However, the survey sample was not representative. We inserted below sentence in method section and limitation section.

We included participants from all region in Japan.

The survey sample was not representative of the structure of Japanese population.

The paper needs to provide an analysis of the reliability and validity of the survey.

Response: Thank you for your insight. The present study had concern on reliability and variability. Regarding reliability, a questionnaire used in previous studies in literature reviews were used to ensure reliability. The answer were given using a Likert scale. However, this study was an online survey and had a small sample size. We should increase sample size and the results should be obtained in face-to-face survey in the future. Regarding validity, the study recruited subjects evenly from all over Japan, however, this study was conducted on population registered with a specific survey company. A representative sample gathering approach should be applied. We inserted above sentences in limitation section.

In Table 1, Basic characteristics of participants, some variables such as Family income, Private income, Occupation are classified too finely, resulting in a small number of items. Suggest merging according to major categories appropriately.

Response: Thank you for your comment. We integrated some items for Family income, Private income, Occupation variables in table1.

Table 3. (base)?B (95% CI)?OR or HR? 

Response: Thank you for your comment. We added explanation in Table 3 and modified Table 3.

Table 3. Multinominal logistic regression analysis of differences between the groups

How was the statistical analysis conducted? What are the independent variables? How are independent variables assigned values? What factors have been verified or controlled? Why only compared gender, age, and comorbidities? Why not compare the reasons for debilitating COVID-19 vaccination?

Response: Thank you for your comment. We inserted below explanation in method section.

“Multinominal logistic regression model was constructed for comparison the participants characteristics between group 2 versus group1, and between group 3 versus group 1. We only enrolled age, sex, and whether participants had comorbidity as independent variable, because the small sample size. “

The conclusion is not clear enough. The main reasons for vaccine hesitancy in different groups should be clearly identified, and corresponding policy recommendations should be proposed.

Response: We appreciate for your suggestion. We mentioned the main reasons for vaccine hesitancy in different groups in discussion section. In addition, we inserted below sentence in discussion section for explanation about the policy recommendations.

“For these participants, educational interventions and seminars by healthcare workers might be effective in sharing necessary information and knowledge about COVID-19 vaccines safety, especially focusing on long-term effects, adverse reactions, and vaccine development history.”

“For these participants, educational interventions and seminars about vaccination safety should be performed with providing information about convenient booking using television, online media, and SNS.”

Pay attention to standardizing chart formats. As shown in Table 1: "Coronavirus was developed by the government as part of a bioweapon program." and Figures 1 and 2, all reasons are followed by a “.”.

Response: Thank you very much for your comment. We standardizing table1, figure 1, and figure2 along with your suggestion.

Reviewer 3 Report

Comments and Suggestions for Authors

This is an important topic and the data and conclusions from this study could be useful for vaccination programmes. There are however some areas that need improved and/or clarified:

Results. Many of the subjects seem to be unemployed (58(22%)). Can this group be described as representative of the population studied (non-vaccinated people in Japan). Is there any data on this?

Results. We are told that 67(26%) had been infected with SARS CoV-2 but how do the authors know if this is accurate or not? many people got infected with few or no symptoms.

table 2 - what is the difference between 'adverse reactions' and 'vaccine safety' - please clarify.

Results. the hierarchical approach established 3 groups based on reasons to decline vaccination. Then the authors performed statistics between the 3 groups (table 2, p<0.05 in all cases). Is this not obvious given that the 3 groups were split up based on these reasons? I'm not following the logic of what additional information table 2 gives us - please clarify and explain your approach here. I have similar comments on the data in table 4.

Table 3. i have no idea what is being presented here. please provide more detail in the table or table legend. What does 'base' mean? what do the square brackets [] mean?

Discussion paragraphs 3 and 4 have large amounts of data in them that don't need to be repeated.

Author Response

This is an important topic and the data and conclusions from this study could be useful for vaccination programmes. There are however some areas that need improved and/or clarified:

Response: Thank you for reviewing our manuscript. We have revised the manuscript in accordance with the comments. The following are our responses to the reviewer’s comments.

Many of the subjects seem to be unemployed (58(22%)). Can this group be described as representative of the population studied (non-vaccinated people in Japan). Is there any data on this?

Response: Thank you for your comment. As you mentioned, the proportion of unemployed was higher than the proportion among general population in Japan. It might associated with the fact that the participants in the study were those who included in the panel of Macromill. We did not have further data on this. We inserted below sentence in limitation section.

the proportion of unemployed was higher than the proportion among general population in Japan.

We are told that 67(26%) had been infected with SARS CoV-2 but how do the authors know if this is accurate or not? many people got infected with few or no symptoms.

Response: Thank you for your insight. We just surveyed the data with self reported questionnaire. We need further assay to identify the correct infection status. We added below in limitation section.

The information on infected status toward COVID-19 were gathered with self-reported questionnaire. We could not identify the population those who were infected without any symptoms.

table 2 - what is the difference between 'adverse reactions' and 'vaccine safety' - please clarify.

Response: Thank you for your comment. We though 'adverse reactions' was more specific, and 'vaccine safety' was more comprehensive cause of hesitation for vaccination.

The hierarchical approach established 3 groups based on reasons to decline vaccination. Then the authors performed statistics between the 3 groups (table 2, p<0.05 in all cases). Is this not obvious given that the 3 groups were split up based on these reasons? I'm not following the logic of what additional information table 2 gives us - please clarify and explain your approach here. I have similar comments on the data in table 4.

Response: Thank you for your comment. Table 2 showed the mean and variance of reasons for declining COVID-19 vaccination with Likert scales each group. As you mentioned, statistical difference was apparently between the 3 groups because the group was divided with the reasons for declining COVID-19 vaccination. However, after careful discussion among authors, we decided to left the statistical result just for conformation and reliability. Table 4 showed the required information about COVID-19 and adequate information sources each 3 group.

Table 3. i have no idea what is being presented here. please provide more detail in the table or table legend. What does 'base' mean? what do the square brackets [] mean?

Response: Thank you for your comment. We modified table3, and inserted explanation about the analysis here.

“Multinominal logistic regression model was constructed for comparison the participants characteristics between group 2 versus group1, and between group 3 versus group 1. We only enrolled age, sex, and whether participants had comorbidity as independent variable, because the small sample size. “

Table 3. Multinominal logistic regression analysis of differences between the groups

Group 1

Group 2

 Group 3

n (%)
(or mean [SD])

Multinominal logistic regression analysis (MLRA)

n (%)
(or mean [SD])

OR (95% CI) in MLRA

p-value in MLRA

n (%)
(or mean [SD])

OR (95% CI) in MLRA

p-value in MLRA

Female

6 (21.4)

(base group)

50 (58.8)

1.64 (0.63 to 2.66)

0.001

70 (47.0)

1.16 (0.19 to 2.12)

0.019

Age

42.5 [13.9]

50.5 [13.5]

0.03 (0.00 to 0.06)

0.047

47.6 [15.9]

0.02 (-0.01 to 0.04)

0.29

With comorbidity

4 (14.3)

27 (31.8)

0.71 (-0.49 to 1.92)

0.25

53 (35.6)

1.03 (-0.12 to 2.18)

0.079

OR=odds ratio; CI=conffidentianl interval; Significant values were in bold.

Discussion paragraphs 3 and 4 have large amounts of data in them that don't need to be repeated.

Response: Thank you for pointing out. We reduced the repetition in discussion paragraphs 3 and 4 adequately.

Round 2

Reviewer 2 Report

Comments and Suggestions for Authors

1.The author answered the reviewer's questions and made modifications to the corresponding sections.

2.Please annotate Group 1, Group 2, and Group 3 under Table 2.

3.Suggest deleting the "Multinomial logistic regression analysis (MLRA)" and "(base group)" below Group 1 in Table 3. Annotation Group 1 is “base group”.

Comments on the Quality of English Language

Minor editing of English language required.

Author Response

The author answered the reviewer's questions and made modifications to the corresponding sections.
Response: Thank you for reviewing our manuscript. We have revised the manuscript in accordance with the comments. The following are our responses to the reviewer’s comments.

Please annotate Group 1, Group 2, and Group 3 under Table 2.

Response: Thank you for your comment. We annotated Group 1, Group 2, and Group 3 under Table 2.

Group 1: group with no specific reason, Group 2: group with clear concerns about trust in the vaccine, Group 3: group with attitudinal barriers and structural barriers.”

Suggest deleting the "Multinomial logistic regression analysis (MLRA)" and "(base group)" below Group 1 in Table 3. Annotation Group 1 is “base group”.

Response: Thank you for your comment. We modified table 3 as your suggestion.

Besides, we improved English writing with English editing service.

Reviewer 3 Report

Comments and Suggestions for Authors

.

Comments on the Quality of English Language

fine

Author Response

Thank you for reviewing our manuscript. We improved English writing with English editing service.